# CROSS MODAL DOMAIN GENERALIZATION FOR QUERY-BASED VIDEO SEGMENTATION

## ABSTRACT

Domain generalization (DG) aims to increase a model's generalization ability against the performance degradation when transferring to the target domains, which has been successfully applied in various visual and natural language tasks. However, DG on multi-modal tasks is still an untouched field. Compared with traditional single-modal DG, the biggest challenge of multi-modal DG is that each modality has to cope with its own domain shift. Directly applying the previous methods will make the generalization direction of the model in each modality inconsistent, resulting in negative effects when the model is migrated to the target domains. Thus in this paper, we explore the scenario of query-based video segmentation to study how to better advance the generalization ability of the model in the multi-modal situation. Considering the information from different modalities often shows consistency, we propose query-guided feature augmentation (QFA) and attention map adaptive instance normalization (AM-AdaIN) modules. Compared with traditional DG models, our method can combine visual and textual modalities together to guide each other for data augmentation and learn a domain-agnostic cross-modal relationship, which is more suitable for multi-modal transfer tasks. Extensive experiments on three query-based video segmentation generalization tasks demonstrate the effectiveness of our method.

## 1 INTRODUCTION

Query-based video segmentation is first introduced by Gavrilyuk et al. (2018), which aims to segment the queried actors or objects in video based on the given natural language query. Although these years have witnessed promising achievements in this field, the segmentation model trained on the source domain will degrade dramatically on the unseen target data due to the domain shift in real applications. As we can see in Figure 1, even if both of the two sentences refer to a standing guy, the visual context complexity, the background environment and the expression styles of the texts in these two cases are quite different, which makes the performance of direct transfer far more unsatisfactory.

Domain adaptation (DA) Kim et al. (2021); Hoffman et al. (2018); Kim et al. (2019) and domain generalization (DG) Dou et al. (2019); Volpi et al. (2018); Choi et al. (2021) have been proposed to solve these problems. Different from DA that requires the acquisition of target domain data during training, which is usually difficult to achieve, DG can learn domain invariant features and improve domain robustness without requiring target-domain data. Previous DG methods use kernel-based optimization to extract domain-agnostic features Muandet et al. (2013); Li et al. (2018c;b), or use meta-learning to simulate domain-shift situations Li et al. (2018a); Liu et al. (2020). However, most of these methods are only suitable for multiple-source domains. Adversarial data augmentation based methods have been proposed to solve the single-domain generalization problem Volpi et al. (2018); Zhao et al. (2020), which use an adversarial loss to generate more realistic fictitious samples as much as possible. However, it is difficult to generate effective meaningful samples that are largely different from the source distribution in semantic segmentation tasks. Thus recently some other methods, such as instance selective whitening Choi et al. (2021) and memory-guided network Kim et al. (2022) have been proposed to handle this problem, and achieve good performance.

Although the above methods have achieved great success, few methods have been proposed specially for multi-modal domain generalization tasks. In query-based video segmentation, domain shift not only exists on the image level (e.g. light, weather, background, etc.) and the instance level (e.g.

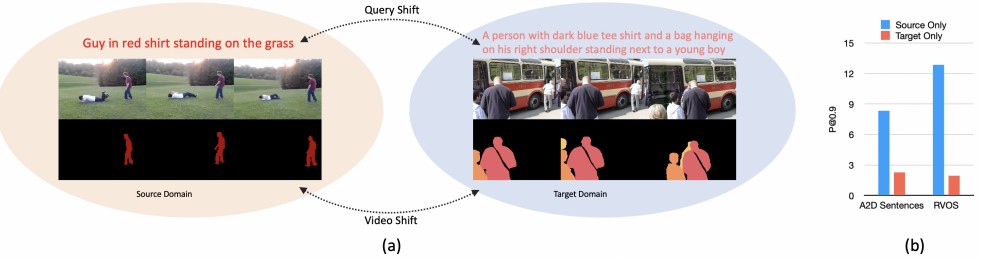

Figure 1: (a) The demo of our multi-modal domain generalization task. (b) The illustration of the necessity of introducing DG in this task. Source only means the model is trained and tested on the same domain, while the target only means the model is trained on another domain, and directly transferred to this domain. As we can see, the performances drop a lot in both two datasets.

size, shape, appearance, etc.)Lin et al. (2021), but also on the natural language level (e.g. expression styles). A natural idea is to gradually augment the source domain data during training to simulate domain shift when facing new domains. However, directly applying the above methods to enhance the two modalities separately and then fuse them together may suffer from negatively affects, since it is difficult to ensure that the generalization directions of these two modalities are consistent.

Thus in this paper, we combine the visual and textual modalities together to facilitate each other for data enhancement. According to our observations, actions belonging to the same type of actors in different domains share similar representations in the query-video latent space, the main reason for the performance degradation is that visual background and contextual complexity vary widely across different domains. Although we can not have access to the target data during training, there exists diverse background information in source data, which can be used to augment the source domain. Therefore, we propose a Query-guided Feature Augmentation (QFA) module, which can use the attention scores between query and video frames to distinguish query-related foreground regions from unrelated background regions. Then we keep the foreground regions to be segmented unchanged, and synthesize novel visual features by gradually enhancing the background areas. To ensure the semantic consistency of the generated data, we introduce Moco-based contrastive learning to force the model to maintain query-related information in the background-perturbed video features. Besides, the expression styles of queries among different domains are different, which will lead to deviations in the attention map between visual and query features when migrating to the target domain. Hence we propose to use AdapIN Huang & Belongie (2017) on the vision-to-query attention map (AM-AdaIN) to alleviate this issue. AdapIN can help the model remove the impact of style in attention map during training, and gradually introduce statistics from other samples to help the model learn robust cross-modal relationships. To be summarized, our main contributions are as follows:

- We are the first to conduct domain generalization on the query-based video-segmentation task, which is also the early attempt to increase generalization ability on cross-modal task.

- We propose two novel QFA and AM-AdaIN modules. Compared with previous DG methods, our model is more suitable for multi-modal generalization task.

- Extensive experiments on three generalization tasks show that our model can greatly enhance the model generalization ability, demonstrating the superiority of our method.

## 2 RELATED WORK

**Domain Generalization.** Domain generalization (DG) requires the model to be robust without accessing the target domain when facing domain shift, which means the model trained only on single or multiple source domains should also perform well on unseen target domain. Early models focus on extracting domain-invariant features Li et al. (2018c;b); Hu et al. (2020) or using kernel-based optimization to minimize the dissimilarity across domains Muandet et al. (2013). Dou et al. (2019) propose a domain-agnostic learning paradigm and encourage the model to learn semantically consistent features across training domains. Huang et al. (2021); Xu et al. (2021) use Fourier-based framework to enhance the domain robustness. Other models aim to enhance the source domains in data-level or feature-level to improve the robustness. Yue et al. (2019) use auxiliary dataset to enhance the source images and obtain different styles of images. Meta-learning is also an important method

to solve generalization problems, which typically partitions source domains into meta-train and meta-test splits to simulate the domain shifts Li et al. (2018a); Liu et al. (2020). As for single domain generalization, which is more challenger than multiple source situation, Volpi et al. (2018); Zhao et al. (2020) use an adversarial data augmentation method which jointly performs domain generalization representation and new domain augmentation in an adversarial learning manner. However, for object or semantic segmentation task, it is difficult for these methods to generate effective fictitious target distribution. Thus instance selective whitening Choi et al. (2021) and memory-guided network Kim et al. (2022) are used to extract domain-invariant features to enhance the generalization capability for semantic segmentation. Although these methods have achieved great success, there are still limit methods target for multi-modal domain generalization problem.

**Query-based Video Segmentation.** Query-based actor-action video segmentation aims to extract relevant regions from videos based on a given natural language query, which is first brought up in Gavrilyuk et al. (2018). Wang et al. (2019) propose an asymmetric cross-guided attention network to incorporate important information from natural language query and visual concepts for each other. Considering the traditional dynamic convolution may neglect spatial context information, Wang et al. (2020) propose a context modulated dynamic convolutional operation, which can integrate natural language query and visual spatial context together to compute the segmentation mask. McIntosh et al. (2020) apply capsule network to encode both the video and textual input, which is more effective than standard convolution based models. Ning et al. (2021) think the spatial relations are also important for this task, thus they propose a polar attention module to make the sentence feature interact with positional embedding more directly. Hui et al. (2021) propose a language-guided feature selection module and a cross-modal adaptive module to select spatial- and temporal- relevant information dynamically and aggregate them comprehensively.

## 3 METHOD

### 3.1 PROBLEM SETUP AND MODEL OVERVIEW

Given a natural language query $Q$ and its counterpart $T$ video frames $V \in R^{T \times H \times W \times C}$, where T, H, W, C are the frame number, height, width and channel number, respectively, the query-based video segmentation aims to generate accurate segment masks on the objects related to the input query. Our new proposed task aims to enhance the generalization ability on unseen target domains $\mathcal{T} \in \{\mathcal{T}_1, ... \mathcal{T}_N\}$ with the segmentation model trained only on single source domain $\mathcal{S}$. The key idea is to improve the model robustness against out-of-distribution shift among different domains. The procedure can be solved as a worst-case problem Sinha et al. (2017); Qiao & Peng (2021):

$$\min_{\phi} \sup_{\mathcal{T}: D(\mathcal{S}, \mathcal{T}) \leq \rho} \mathbb{E}[\mathcal{L}_{\text{seg}}(\phi; \mathcal{T})], \tag{1}$$

where $\phi$ are the model parameters for segmentation objective function $\mathcal{L}_{\text{seg}}$, D is a similarity metric to measure domain distance between $\mathcal{S}$ and $\mathcal{T}$, $\rho$ is the largest domain distance.

Considering it is difficult to directly optimize the worst-case problem, we try to solve the query-based video segmentation generalization task by gradually generating a novel domain $\bar{\mathcal{S}}$ with our proposed QFA3.3 and AM-AdaIN3.4 during training, and $D(\mathcal{S}, \bar{\mathcal{S}}) \leq \rho$. Besides, we apply moco-based contrastive learning to constrain the distance $\leq \rho$.

### 3.2 BASELINE MODEL

First we use an I3D layer Carreira & Zisserman (2017) with stacked 3D convolution as our visual encoder and get visual representations as $V \in R^{T \times H \times W \times C}$, then we apply average pooling over the temporal dimension to get $\hat{V}$. Since the usage of vocabulary in different datasets is different, to enhance the generalization ability and learn some words with similar semantics but different expressions, we choose Bert as our text Encoder, the natural language query can be encoded as $Q \in R^{L \times C}$, where $L$ is the length of the query. Followed by Hui et al. (2021), we divide the visual features in the encoder and decoder layers into K different scales respectively, denoted as $\hat{V}_k \in R^{H_k \times W_k \times C}$, where $H_k$, $W_k$ and $C$ are height, width and channel number of the i-th visual feature, respectively. To allow the visual features interacting with the query more comprehensively, we design a query-vision interaction module(QVIM).

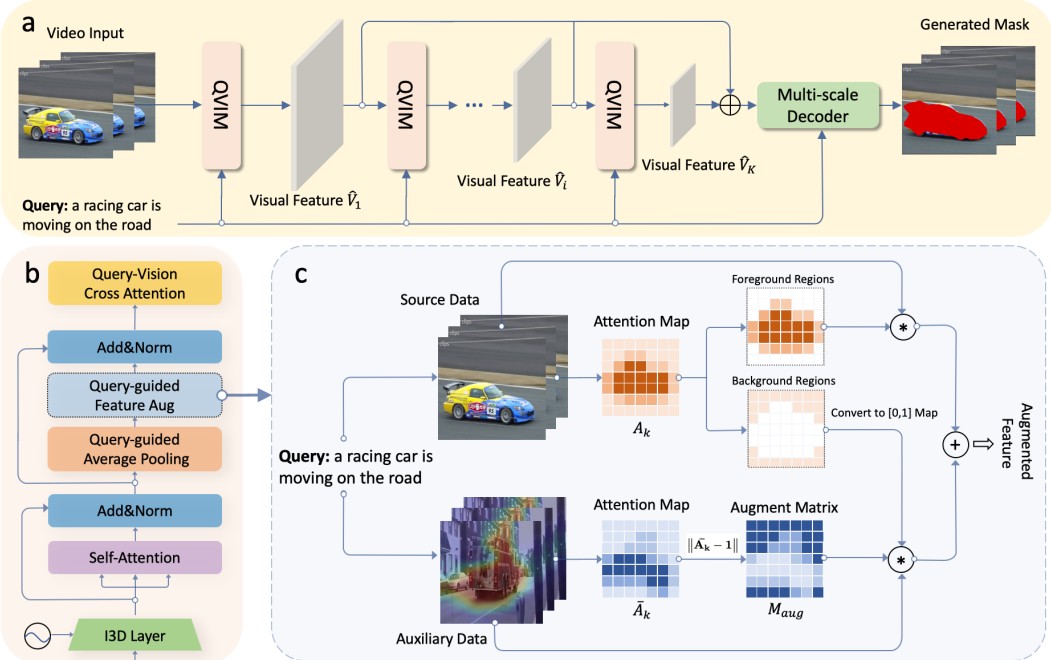

Figure 2: The overview of our proposed network. (a) The main pipeline of our model. (b) The architecture of query-vision interaction module (QVIM), notice the query-guided feature augmentation (QFA) only exists in one layer. (c) Our proposed QFA first uses the query to distinguish the query-related foreground and unrelated background regions in source video frames, then selects query-agnostic regions from auxiliary data. Finally we use the query-agnostic regions to enhance the background areas of the original visual features and obtain the augmented data. The $M_{aug}$ can avoid the interference of similar objects (e.g., truck) in the auxiliary data to the source data.

**Query-Vision Interaction Module** Our QVIM aims to enhance the interaction between different scale visual features and natural language. Followed by Wang et al. (2019), we adopt an 8-dimensional position encoding feature to encode relative position information of each pixel. Then we apply a self-attention layer on the visual feature to enable each pixel can perceive context-related information. Chen et al. (2021) have proved that attentional pooling can help model localize its attention to semantically more important regions, which can further enhance domain generalization ability. Thus in this task, we employ query vectors to obtain relevant spatial regions from visual features. The k-th layer attention scores $A_k$ between query and each visual pixels can be defined as:

$$a_{i,j}^k = \frac{\langle v_{i,j}^k, q \rangle}{\sum_{i',j'} \langle v_{i',j'}^k, q \rangle} (i' \in [1, H_k], j' \in [1, W_k]), \tag{2}$$

where $q \in R^C$ is the RNN output from Q. We use $A_k$ to highlight the most query-related regions from input visual features, which can facilitate the subsequent segmentation task. Besides, we apply another cross-modality attention in Eq 7 between visual inputs and natural language query, which can select important information from query to enhance the representation of each pixel.

**Multi-scale Decoder Layer** Followed by Wang et al. (2019), we divide the visual features into K different scales, each scale is obtained by a I3D and a QVIM layer. Then we combine average-pooling and fully convolutional layer to generate multi-scale response map $\{\{s_{i,j}^k\}_{i=1,j=1}^{H_i^k \times W_j^k}\}(k \in [1, K])$ from the encoded visual features $\hat{V}_k \in R^{H_k \times W_k \times C}$.

### 3.3 QUERY-GUIDED FEATURE AUGMENTATION

Data augmentation is an important technique to improve model generalization ability. However, previous works are not suitable for multi-modal generalization tasks. For example, Fourier-based methods Huang et al. (2021); Xu et al. (2021) will cause the color of vital objects changed, conflicting

with the colors described in the question text. Also, it is difficult for adversarial-based methods Volpi et al. (2018); Zhao et al. (2020); Qiao et al. (2020) to synthesize meaningful fictitious video frames in segmentation task. Thus in this section, we take advantage of the consistency of information in multi-modality, and use the query as guidance to gradually make meaningful enhancements to the source domain in feature level. Concretely, given an arbitrary scale of visual features $\hat{V}_k \in R^{H_k \times W_k \times C}$, we distinguish important foreground regions $R_f$ from background regions $R_b$ according to the attention score $a_{i,j}^k$. We set a threshold $\beta$, the pixels larger than $\beta$ are considered as foreground regions $R_f$ which is highly relevant to the question, while the others smaller than $\beta$ are considered as irrelevant background parts $R_b$:

$$\mathbf{A_k} = \begin{cases} R_f, & \text{if } a_{i,j}^k > \beta \\ R_b, & \text{if } a_{i,j}^k <= \beta \end{cases} \quad i' \in [1, H_k], j' \in [1, W_k], \tag{3}$$

then we convert $R_b$ into a Bool matrix $B$, the pixels with values are set to 1, and the pixels without values are set to 0. Next, we randomly select another video from the same batch as auxiliary data $\hat{V}_k^{au}$. Similarly, we use the query and the auxiliary data to calculate the attention scores $\bar{A}_k$, and then subtract the attention matrix from the all-one matrix to get the augment matrix $M_{aug}$:

$$M_{aug} = \|\bar{\mathbf{A_k}} - \mathbf{1}\| \tag{4}$$

The purpose of Eq 4 is to minimize the interference of query-related objects in auxiliary data to important regions of the original visual features. Finally, we combine the auxiliary background and the original visual features together to generate new augmented data $\tilde{V}_k$:

$$\tilde{V}_k = B * M_{aug} * \hat{V}_k^{au} + R_f * \hat{V}_k \tag{5}$$

At the test time, we remove the QFA module and directly conduct inference on original visual features from target domains.

**Progressive Manner:** In order to make the query-guided feature augmentation procedure more robust, here we adopt a progressive expansion manner. At the initial few epochs, we do not perform any data augmentation, allowing the model to learn a stable attention relationship between query and visual features. Then we gradually shift the threshold $\beta$ from small to large during the training process, which means the replaced background area in the original video frames is progressively getting larger. To ensure that the augmented data still retains the key region features relevant to its counterpart query, we then introduce a semantic constraint objective function.

**Constraints on Augmented Data:** Previous works have proved that it is necessary to apply distance constraints between source domain and augmented domains Qiao et al. (2020); Qiao & Peng (2021), or using stochastic gradients to update the generated adversarial examples Volpi et al. (2018); Zhao et al. (2020). While in this paper, we employ contrastive learning to pull the distance between query and its counterpart augmented visual features, and push the distance of other mismatched data. To increase the diversity of negative samples, we follow Moco He et al. (2020), by designing a dynamic dictionary with a queue to store negative samples. Given a query as q, its counterpart augmented visual data as positive key $k_+$, and $N - 1$ samples from the dictionary as negative keys, we apply InfoNCE loss Van den Oord et al. (2018) to measure the similarities between q and keys:

$$\mathcal{L}_{\text{const}} = -log \frac{exp(q \cdot k_+ / \tau)}{\sum_{n=0}^{N} exp(q \cdot k_n / \tau)}, \tag{6}$$

where $\tau$ is a temperature hyper-parameter. As the training progresses, the magnitude of data enhancement is gradually increasing, and the ability of $\mathcal{L}_{\text{const}}$ to measure the distance between the query and positive/negative samples also improves accordingly. Compared with adversarial learning, our model can converge more stably, and our model can measure the semantic similarity between cross-modalities better than Wasserstein distance Qiao et al. (2020).

### 3.4 ADAPTIVE INSTANCE NORMALIZATION ON ATTENTION MAP

From Figure 1 we can see that the domain shift not only exists in visual level, but also exists in natural language description style. This will lead to a deviation in the cross-attention map between the visual features and the query text when the model is migrated to the target domain. The model cannot pay attention to the key text positions, thus the unimportant text information is integrated into

the visual pixels' representation, which will cause deviations in the segmentation results. To bridge this gap, we introduce style randomization on the cross attention map $C_k$ between $V_k \in R^{H_k W_k \times C}$ and $Q \in R^{L \times C}$, which can be computed as:

$$C_k = softmax(\delta(W(V_k \cdot Q^T))) \in R^{H_k \times W_k \times L}, \tag{7}$$

$$\mu(C_k) = \frac{1}{H_k W_k} \sum_{h=1}^{H_k} \sum_{w=1}^{W_k} c_{hw}^k \in R^L, \quad \sigma(C_k) = \sqrt{\frac{1}{H_k W_k} \sum_{h=1}^{H_k} \sum_{w=1}^{W_k} (\mathbf{c}_{hw}^k - \mu(C_k))^2 + \epsilon}, \tag{8}$$

where W is a learning parameter, $\delta$ is the Relu activation function, $\epsilon$ is a small number to avoid zero. After obtaining the feature statistics of source cross attention map $C_k$, we can use adaptive instance normalization (AdaIN) Huang & Belongie (2017) to replace the statics of input with an arbitrary chosen $C_k^{'}$:

$$AdaIN(C_k, C_k^{'}) = \hat{\sigma}(\frac{C_k - \mu(C_k)}{\sigma(C_k)}) + \hat{\mu}, \tag{9}$$

$$\hat{\mu} = (1 - \alpha) \cdot \mu(C_k) + \alpha \cdot \mu(C_k^{'}), \quad \hat{\sigma} = (1 - \alpha) \cdot \sigma(C_k) + \alpha \cdot \sigma(C_k^{'}), \tag{10}$$

where $\alpha \sim U(0, 0.5)$. By introducing the mean and variance from other $C_k^{'}$ as perturbed data, the model can learn domain-invariant response regions from the visual-to-query attention map, which can extract important information from natural language query against expression style shift when transferring to target domains. During inference time, we also remove the AdapIN layer.

## 3.5 TRAINING

Here we introduce how to incorporate our proposed QFA and AM-AdaIN modules into the baseline model. Although these two modules can be inserted into query-guided average pooling and query-vision cross attention in QVIM layer at any scale, respectively. Inserting into different layers will have different effects on the generalization ability of the model. We find that data augmentation is suitable for the largest scale of visual features, while AM-AdaIN is suitable for the smaller scale, the details will be analyzed in section 4.4.

After obtaining the multi-scale response map $\{\{s_{i,j}^k\}_{i=1,j=1}^{H_i^k \times W_j^k}\}(k \in [1, K])$ in Section 3.2, and the ground-truth pixel-level annotations $\{\{y_{i,j}^k\}_{i=1,j=1}^{H_i^k \times W_j^k}\}(k \in [1, K]), y_{i,j}^k \in \{0, 1\})$, we can compute the multi-scale segment loss with binary cross-entropy loss:

$$\mathcal{L}_{seg} = -\frac{1}{K} \frac{1}{H_k W_k} \sum_{h=1}^{H_k} \sum_{w=1}^{W_k} (y_{i,j} \cdot log(s_{i,j}) + (1 - y_{i,j}) \cdot log(1 - s_{i,j})) \tag{11}$$

The total loss function can be computed as $\mathcal{L} = \mathcal{L}_{seg} + \mathcal{L}_{const}$.

## 4 EXPERIMENTS

### 4.1 DATASETS AND GENERALIZATION TASKS

**A2D Sentences** is first released by Gavrilyuk et al. (2018), they provide corresponding natural language descriptions for each video in Actor-Action Dataset Xu et al. (2015). It has a total of 3,782 videos, which contains 8 action classes (e.g. walking, eating, running) and 7 actor classes (e.g. adult, dog, cat). **Refer-Youtube-VOS (RVOS)** is first extended on the Youtube-VOS dataset Xu et al. (2018) by Seo et al. (2020), which contains 3975 high-resolution videos with 94 common object categories. RVOS is annotated with 27899 text expressions. **J-HMDB Sentences** is also brought up by Gavrilyuk et al. (2018) from Jhuang et al. (2013), which comprises 928 videos with 21 action classes, annotated with 928 sentences.

**Generalization Tasks:** In this paper, we evaluate our methods on three generalization tasks: (1) **A2D Sentences**⟶**RVOS**$_{part}$($A2R$) : since the RVOS contains overall 94 object categories, we select the same video object types as A2D, e.g. car, person, dog, cat, with a total of 3528 video-query pairs. The domain generalization task requires that the types of objects tested on the target domain should exist in the source domain. (2) **RVOS**⟶**A2D Sentences**($R2A$) : the types of objects in

Table 1: Comparison with state-of-the-art methods on three generalization tasks.

| Method | Precision | | | | | mAP | IoU | |
|---|---|---|---|---|---|---|---|---|
| | P@0.5 | P@0.6 | P@0.7 | P@0.8 | P@0.9 | 0.5:0.95 | Overall | Mean |
| A2R | | | | | | | | |
| Baseline | 38.46 | 30.61 | 21.00 | 10.29 | 1.96 | 18.54 | 40.66 | 35.36 |
| SFA | 39.17 | 32.26 | 22.56 | 10.97 | 2.18 | 19.37 | 40.74 | 35.64 |
| AdaIN | 40.31 | 32.71 | 23.16 | 12.24 | 2.61 | 20.12 | 42.78 | 36.57 |
| RobustNet | 40.90 | 33.02 | 23.30 | 11.99 | 2.60 | 20.56 | 43.21 | 37.26 |
| Mixstyle | 41.21 | 33.45 | 23.44 | 12.41 | 2.73 | 20.89 | 43.01 | 37.85 |
| DSU | 41.58 | 34.16 | 24.38 | 12.95 | 2.89 | 20.93 | 43.11 | 38.09 |
| Ours | **42.62** | **35.13** | **25.45** | **13.38** | **3.29** | **21.36** | **44.06** | **38.31** |
| R2A | | | | | | | | |
| Baseline | 48.65 | 40.95 | 30.27 | 16.11 | 2.29 | 25.12 | 52.17 | 43.81 |
| SFA | 49.96 | 42.65 | 31.63 | 17.23 | 2.31 | 26.06 | 51.76 | 43.76 |
| AdaIN | 50.19 | 43.50 | 33.05 | 17.53 | 2.60 | 26.78 | 52.32 | 43.98 |
| DSU | 51.20 | 43.14 | 32.41 | 17.50 | 2.86 | 26.79 | 53.14 | 44.50 |
| Mixstyle | 50.50 | 42.50 | 32.59 | 19.02 | 3.27 | 27.12 | 53.27 | 44.23 |
| RobustNet | 51.64 | 44.30 | 35.93 | 18.22 | 2.52 | 27.35 | 53.08 | 44.55 |
| Ours | **53.02** | **45.87** | **35.93** | **20.69** | **3.66** | **29.09** | **54.73** | **46.05** |
| A2J | | | | | | | | |
| Baseline | 83.92 | 70.66 | 43.68 | 10.10 | 0.10 | 37.44 | 63.74 | 63.34 |
| SFA | 85.88 | 73.14 | 46.49 | 11.37 | 0.13 | 38.97 | 65.21 | 64.32 |
| AdaIN | 86.39 | 73.42 | 46.98 | 12.63 | 0.13 | 39.50 | 66.04 | 65.89 |
| RobustNet | 87.01 | 74.25 | 46.57 | 10.83 | 0.15 | 39.27 | 66.14 | 65.09 |
| DSU | 88.15 | 74.79 | 48.33 | 11.69 | 0.16 | 40.07 | 66.29 | 65.76 |
| Mixstyle | 87.91 | 75.48 | 49.50 | 12.65 | 0.17 | 40.58 | 67.20 | 65.62 |
| Ours | **89.25** | **76.17** | **49.86** | **12.90** | **0.20** | **41.23** | **67.87** | **67.19** |

A2D Sentences are included in RVOS, so the model can be directly migrated to A2D Sentences after training on RVOS. (3) **A2D Sentences→J-HMDB Sentences**($A2J$) : following previous works Gavrilyuk et al. (2018); Wang et al. (2019; 2020), we also use the model trained on A2D Sentences to evaluate the generalization ability on J-HMDB Sentences without any additional fine-tuning.

## 4.2 BASELINES AND EVALUATION METRICS

We use the multi-layer QVIM encoder and decoder as our baseline model without any consideration of domain generalization. We also compare our method with five state-of-the-art DG models: Simple Feature Augmentation (SFA)Li et al. (2021), Adaptive Instance Normalization (AdaIN) Huang & Belongie (2017), Instance Selective Whitening (RobustNet) Choi et al. (2021), Mixstyle Zhou et al. (2020), Domain Shifts with Uncertainty (DSU) Li et al. (2022). Following previous works, we adopt intersection-over-union (IoU) to measure the model segmentation ability, more details can be found in appendix A.1. The implementation details can be found in appendix A.2.

## 4.3 MAIN RESULTS

First we compare our full model with the existing domain generalization methods to verify the effectiveness of our proposed methods. We reimplement these state-of-the-art DG modules on our baseline model. The main evaluation results on three generalization tasks are presented in Table 1. From the results we can see that our method outperforms the baseline model by a large margin on all three generalization tasks, demonstrating the effectiveness of our model. Compared with other traditional DG methods that only consider single-modal domain enhancement, our model can combine vision and text modalities with each other to perform guided data enhancement respectively, which is more suitable for situations when both two modalities suffer from domain shift. Besides, the results in Table 1 and 2 indicate that our QFA can achieve better performance than AdaIN Huang & Belongie (2017) and Mixstyle Zhou et al. (2020), demonstrating it is necessary to use natural language query to guide the visual feature augmentation.

Table 2: Analysis of the components on two generalization tasks.

| Method | Precision | | | | | mAP | IoU | |
|---|---|---|---|---|---|---|---|---|
| | P@0.5 | P@0.6 | P@0.7 | P@0.8 | P@0.9 | 0.5:0.95 | Overall | Mean |
| | A2R | | | | | | | |
| Our Full Model | 42.62 | 35.13 | 25.45 | 13.38 | 3.29 | 21.36 | 44.06 | 38.31 |
| w/o QGAP | 40.82 | 33.05 | 24.01 | 11.85 | 2.64 | 20.40 | 42.41 | 36.85 |
| w/o cross-att | 39.37 | 31.89 | 22.39 | 11.82 | 2.58 | 19.44 | 41.57 | 35.87 |
| w/o $\mathcal{L}_{\mathrm{const}}$ | 41.58 | 34.16 | 24.38 | 12.95 | 2.88 | 20.73 | 43.14 | 37.29 |
| w/o QFA | 40.16 | 32.00 | 21.97 | 11.03 | 2.24 | 19.39 | 41.76 | 36.35 |
| w/o AM-AdaIN | 41.72 | 33.59 | 24.23 | 12.81 | 2.66 | 20.88 | 42.64 | 37.67 |
| | R2A | | | | | | | |
| Our Full Model | 53.02 | 45.87 | 35.93 | 20.69 | 3.66 | 29.09 | 54.73 | 46.05 |
| w/o QGAP | 49.96 | 42.65 | 31.63 | 17.23 | 2.31 | 26.06 | 51.76 | 43.76 |
| w/o cross-att | 49.21 | 42.16 | 32.56 | 17.86 | 2.18 | 25.86 | 51.22 | 43.15 |
| w/o $\mathcal{L}_{\mathrm{const}}$ | 52.10 | 44.38 | 33.28 | 18.46 | 3.14 | 27.54 | 53.10 | 44.88 |
| w/o QFA | 50.45 | 43.55 | 33.56 | 18.58 | 2.83 | 27.33 | 53.15 | 44.52 |
| w/o AM-AdaIN | 51.76 | 44.27 | 33.87 | 18.71 | 3.35 | 27.72 | 53.31 | 45.09 |

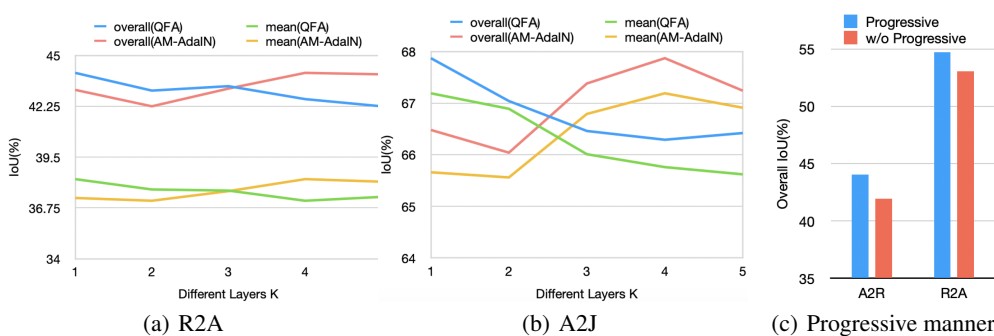

(a) R2A        (b) A2J        (c) Progressive manner

Figure 3: Analysis of the QFA and AM-AdaIN in different layers (a) and (b). Analysis of the progressive manner (c).

## 4.4 ABLATION STUDIES

**Model Components** To investigate the effect of individual components, we remove them from the full model and re-evaluate them on two generalization tasks, as shown in Table 2. To demonstrate the effectiveness of the query-vision interaction module (QVIM), we remove the query-guided average pooling (QGAP) and cross-attention module respectively, denotes as "w/o QGAP" and "w/o cross-att". As we can see, the generalization performance drops a lot on both two tasks, which shows that the multi-layer QVIM plays an essential role in this task. To further verify our proposed query-guided feature augmentation (QFA) and whether it is necessary to apply constraints on augmented data, we remove QFA and $\mathcal{L}_{\mathrm{const}}$, denotes as "w/o QFA" and "w/o $\mathcal{L}_{\mathrm{const}}$". The results show that QFA can greatly advance the generalization ability of the model. Also without $\mathcal{L}_{\mathrm{const}}$ the performance drops, demonstrating the constraints on augmented data can guarantee the model generates more meaningful and query-preserving augmentations. Meanwhile, the AM-AdaIN can also improve the model's robustness by learning a domain-agnostic vision-query relationship during training.

**The Effect of QFA And AM-AdaIN in Different Layers** We find through experiments that multi-layer QFA and AM-AdaIN will greatly reduce the performance of the model, thus we insert the two modules into only one QVIM layer respectively. In this subsection, we will study the effect of these two modules on different scales. From the Figure 3 (a) and (b) we can see that QFA has the best performance improvements on the model when K=1, and the worst when K=5. AM-AdaIN has the best performance improvements when K=4. The results demonstrate the QFA is more suitable for

Table 3: Analysis of the bool matrix and augment matrix used in QFA module.

| Method | A2R | | | R2A | | | A2J | | |
|---|---|---|---|---|---|---|---|---|---|
| | mAP | IoU | | mAP | IoU | | mAP | IoU | |
| | 0.5:0.95 | Overall | Mean | 0.5:0.95 | Overall | Mean | 0.5:0.95 | Overall | Mean |
| w/o Bool Matrix | 19.14 | 41.20 | 36.38 | 25.50 | 52.28 | 43.09 | 39.15 | 65.78 | 65.32 |
| w/o Augment Matrix | 20.12 | 42.78 | 36.97 | 26.81 | 53.24 | 44.76 | 39.64 | 66.21 | 65.79 |
| *Full Model* | 21.36 | **44.06** | **38.31** | 29.09 | **54.73** | **46.05** | **41.23** | **67.87** | **67.19** |

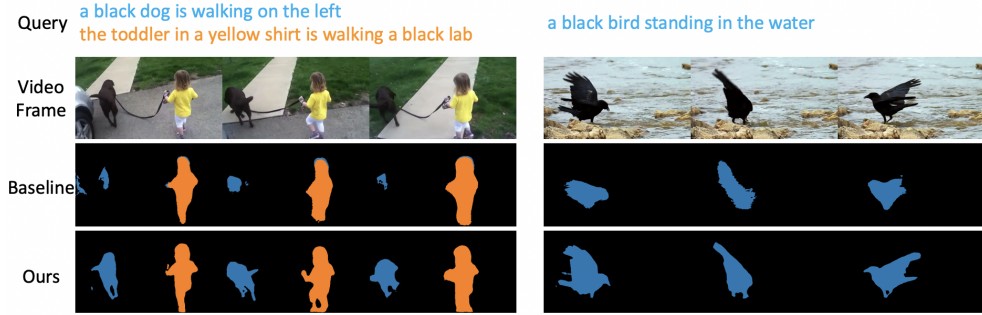

Figure 4: The visualization of segmentation results. The left is the result on A2D Sentences of the R2A task. The right is the result on the RVOS of the A2R task.

enhancing visual features at larger scales, while AM-AdaIN is more suitable for deeper interactions between visual features and query representations.

**The Effect of the Components in QFA** To investigate the effectiveness of two matrices used in Eq 3.3, we remove the bool matrix $B$ and augment matrix $M_{aug}$ respectively to test the model's performance on three tasks, as depicted in Table 3. The results show that without bool matrix $B$, the performance will drop a lot, which demonstrate it is important to use the bool matrix to keep the query-related regions in original visual features unchanged. The introduce of augment matrix can also guarantee that the query-related objects in the auxiliary data will not interfere the original video features, which is important for QFA module. Also we test whether it is necessary to apply progressive manner in data augmentation, as shown in Figure 3(c). The results illustrate without progressive manner, it is difficult for the model to learn meaningful data augmentation, which will deteriorate the transferring performance.

## 4.5 QUALITATIVE ANALYSIS

To further qualitatively compare our method with baseline model, we visualized two segmentation results, as shown in Fig 4. From the results we can see that compared with baseline model, the introduce of QFA and AM-AdaIN can help the model distinguish the query-related objects from the background more accurately. Notice that in the first demo, the baseline model can barely distinguish the dog from its surrounding objects. However, our full model can continuously replace the surrounding background regions during training, allowing the model to learn the difference between the objects and the surrounding environment more robustly. More qualitative results can be found in appendix material 5 6 7.

## 5 CONCLUSION

In this paper, we introduce a new task for domain generalization, generalizable query-based video segmentation, which aims to train a model on the source domain that can segment video objects according to the query text, and can also generalize to the unseen target domain. Previous methods are seldom designed for multi-modality tasks, especially when each modality has its own domain shift. To bridge this gap, we propose QFA and AM-AdaIN modules, which can combine the text and vision modalities together to increase the holistic generalization ability of the model. We conduct extensive experiments on three datasets, and the results demonstrate that our methods can effectively promote the multi-modal domain generalization task.

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

Table 4: Analysis of the parameter numbers of our proposed modules.

|   | Baseline | QFA | $\mathcal{L}_{\text{const}}$ | AM-AdaIN | Parameters Numbers |
|---|---|---|---|---|---|
| 1 | $\checkmark$ | | | | 109.71G |
| 2 | $\checkmark$ | $\checkmark$ | | | 110.26G |
| 3 | $\checkmark$ | $\checkmark$ | $\checkmark$ | | 112.36G |
| 4 | $\checkmark$ | $\checkmark$ | $\checkmark$ | $\checkmark$ | 112.36G |

Riccardo Volpi, Hongseok Namkoong, Ozan Sener, John C Duchi, Vittorio Murino, and Silvio Savarese. Generalizing to unseen domains via adversarial data augmentation. *Advances in neural information processing systems*, 31, 2018.

Hao Wang, Cheng Deng, Junchi Yan, and Dacheng Tao. Asymmetric cross-guided attention network for actor and action video segmentation from natural language query. In *Proceedings of the IEEE/CVF International Conference on Computer Vision*, pp. 3939–3948, 2019.

Hao Wang, Cheng Deng, Fan Ma, and Yi Yang. Context modulated dynamic networks for actor and action video segmentation with language queries. In *Proceedings of the AAAI Conference on Artificial Intelligence*, volume 34, pp. 12152–12159, 2020.

Chenliang Xu, Shao-Hang Hsieh, Caiming Xiong, and Jason J Corso. Can humans fly? action understanding with multiple classes of actors. In *Proceedings of the IEEE Conference on Computer Vision and Pattern Recognition*, pp. 2264–2273, 2015.

Ning Xu, Linjie Yang, Yuchen Fan, Jianchao Yang, Dingcheng Yue, Yuchen Liang, Brian Price, Scott Cohen, and Thomas Huang. Youtube-vos: Sequence-to-sequence video object segmentation. In *Proceedings of the European conference on computer vision (ECCV)*, pp. 585–601, 2018.

Qinwei Xu, Ruipeng Zhang, Ya Zhang, Yanfeng Wang, and Qi Tian. A fourier-based framework for domain generalization. In *Proceedings of the IEEE/CVF Conference on Computer Vision and Pattern Recognition*, pp. 14383–14392, 2021.

Xiangyu Yue, Yang Zhang, Sicheng Zhao, Alberto Sangiovanni-Vincentelli, Kurt Keutzer, and Boqing Gong. Domain randomization and pyramid consistency: Simulation-to-real generalization without accessing target domain data. In *Proceedings of the IEEE/CVF International Conference on Computer Vision*, pp. 2100–2110, 2019.

Long Zhao, Ting Liu, Xi Peng, and Dimitris Metaxas. Maximum-entropy adversarial data augmentation for improved generalization and robustness. *Advances in Neural Information Processing Systems*, 33:14435–14447, 2020.

Kaiyang Zhou, Yongxin Yang, Yu Qiao, and Tao Xiang. Domain generalization with mixstyle. In *International Conference on Learning Representations*, 2020.

# A APPENDIX

## A.1 EVALUATION METRICS

For IoU, we use the "Overall IoU", which calculates the ratio of the total intersection area divided by the total union area over the entire dataset, and the "Mean IoU", which first calculates the ratio of each sample and then obtains the average results on the whole dataset. "P@K" denotes compared with ground truth results, the IoU scores of testing samples are larger than K. We also measure the mean average precision at 5 different IoU thresholds from 0.50 to 0.95 with the step 0.05.

## A.2 IMPLEMENTATION DETAILS

For natural language query inputs, we set the maximum number of words in one query as 20, and apply the BertDevlin et al. (2018) as our text encoder. For video inputs, we employ the I3D network Carreira & Zisserman (2017) pretrained on the Kinetics dataset to extract the spatial and

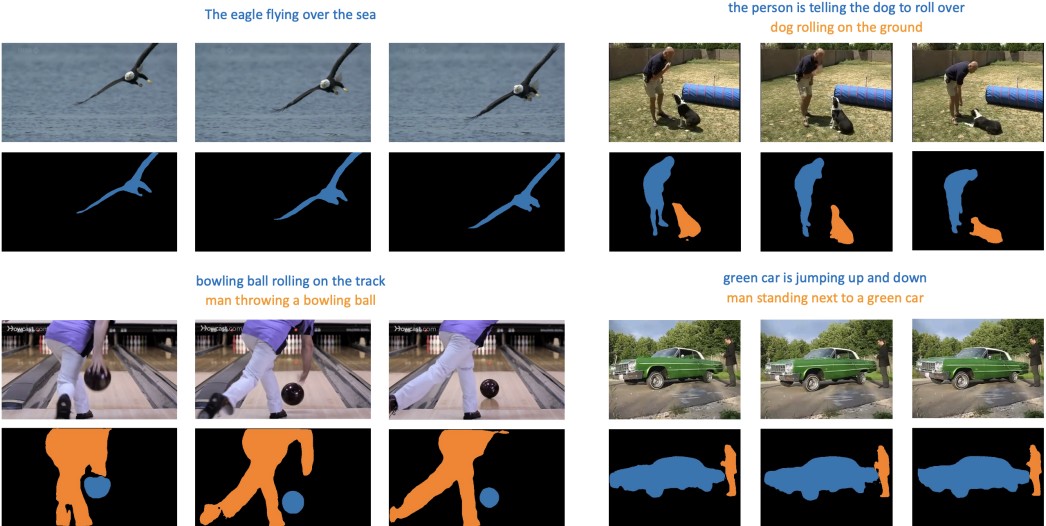

Figure 5: The visualization results of generalization task:**RVOS**⟶**A2D Sentences**.

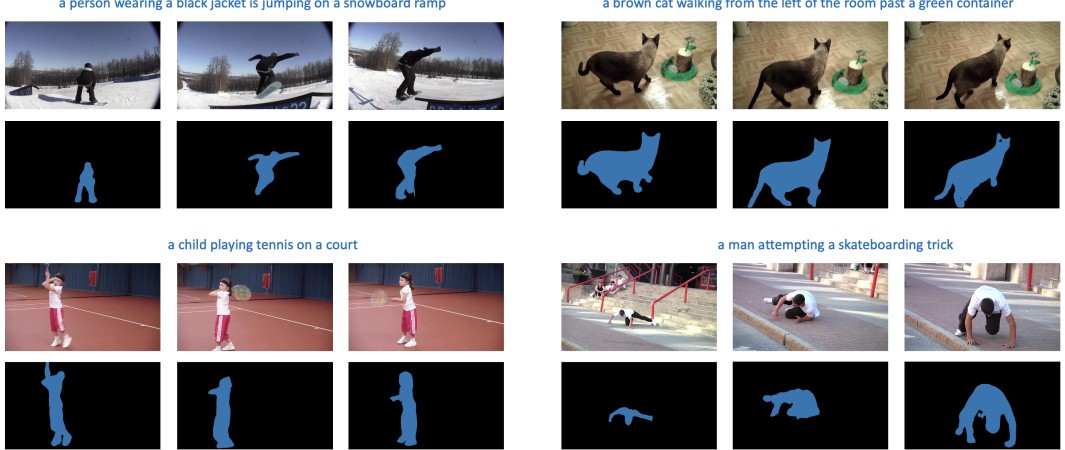

Figure 6: The visualization results of generalization task:**A2D Sentences**⟶**RVOS**$_{part}$.

temporal features and we use the pre-trained ResNet-50He et al. (2016) to extract each video frame representations, the number of frames in one clip is 8. We divide the visual features into $K = 5$ different scales, the sizes of them are $320 \times 320$, $160 \times 160$, $80 \times 80$, $40 \times 40$ and $20 \times 20$ separately. We set the hidden size of visual and query features as 512. Following Wang et al. (2019), the FCN network in deconvolutional layer contains three fully convolutional layers, where the kernel size is 3×3 for the first two layers and 1×1 for the remaining layer. For first 5 training epochs, we do not apply data augmentation and AdaIN in our baseline model, then we gradually increase the threshold $\beta$ from 0.05 to 0.30 by 0.05 every 10 training epochs. All experiments are implemented with Pytorch package on 4 NVIDIA V100 GPUs in this paper, the batch size is 16, and we use Adam optimizer with a initial learning rate 1e-7.

For the implementation of other baseline methods, since the previous DG methods are all designed especially for single modality, here we implement these models on visual modality to test their performances, because we found that deploying them in visual modality works better than text modality. And if they are deployed in both two modalities, the performances are even worse.

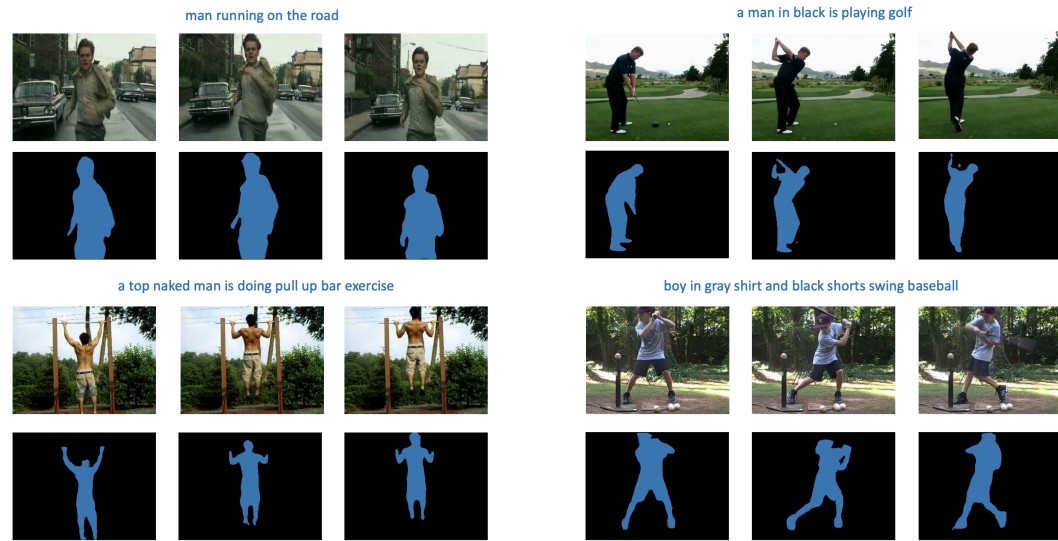

Figure 7: The visualization results of generalization task:**A2D Sentences**⟶**J-HMDB Sentences**.

### A.3 DISCUSSIONS

The above experiments illustrate our proposed methods can effectively optimize the worst-case problem by generating a novel domain $\bar{\mathcal{S}}$. At the initial training, $\bar{\mathcal{S}}$ is close to original domain $\mathcal{S}$. As the training progresses, $\beta$ is gradually increasing, thus more background areas will be enhanced, the distance $D\left(\mathcal{S}, \bar{\mathcal{S}}\right)$ is also getting larger. At the same time, the $\mathcal{L}_{\text{const}}$ can guarantee the generated data has similar query-related regions to source data. Also our AM-AdaIN can introduce style randomization on cross-attention map to make the model learn a more robust relationship between query and visual feature. Furthermore, our methods consume very limited computing resources compared with the baseline model, the details can be found in the next subsection.

### A.4 MODEL PARAMETERS

Our proposed modules only consume very limit computing resources compared with the baseline model, as depicted in Table 4. The increases of parameters of QFA, $\mathcal{L}_{\text{const}}$ and AM-AdaIN are 0.55G, 2.1G, 0 respectively. Compared with traditional DG methods such as meta-learning and adversarial data augmentation, our model can effectively reduce parameters and can be easily deployed to multi-modal generalization tasks.

