# OpenReview forum: "Cross Modal Domain Generalization for Query-based Video Segmentation"
_ICLR.cc/2023/Conference — Submitted to ICLR 2023_

### Official Review · Reviewer_XKxN · 2022-10-23

**Confidence:** 3
**Correctness:** 3
**Technical Novelty And Significance:** 3
**Empirical Novelty And Significance:** Not applicable
**Recommendation:** 3

**Clarity, Quality, Novelty And Reproducibility:**

Clarity: Needs Improvement.
Quality: Needs Improvement.
Novelty: Somewhat novel.
Reproducibility: Reproducible.


**Strength And Weaknesses:**

Strength
+ A good attempt to improve domain generalization of the query-based video segmentation methods.
+ The experimental results show that the proposed method outperforms the competing methods.

Weaknesses
+ This work proposes to augment training data with the proposed QFA. Only the background information is replaced during the augmentation. However, the foreground information also varies significantly for complex environments. For instance, as shown in Figure 1, the targets in the source and target domains have different scales.
+ I am still confused about the detailed motivation for using AdaIN. It is originally used in style transfer, and this work seems to use it to mitigate the “style” of the query. However, AdaIN requires supervised training to present different styles, while the target domain data is not available during training for this task.
+ The technical novelty of this paper is not up to the standard of ICLR. I believe that model performance is (marginally) improved but combining a data augmentation method and AdaIN and then applying them to the segmentation task is not appealing to me.
+ The presentation should be improved. For instance, in the first sentence of the introduction, it seems that “byGavrilyuk” should be separated into two worlds by spacing. Similarly, “AdapINHuang” in the last paragraph of the introduction.


**Summary Of The Paper:**

This work aims at improving query-based video segmentation based on domain generalization. Specifically, this work focuses on solving the challenges of cross-modal settings, where domain shift exists at both the text and visual levels. To mitigate these challenges, this work proposes the query-guided feature augmentation method and uses AdaIN to modulate the text query.

**Summary Of The Review:**

This work views query-based segmentation as a cross-domain generalization problem. It proposes a data augmentation scheme and uses AdaIN to improve model performance. The technical novelty of this paper is not up to the standard, and the motivations are not clear enough. In addition, the presentation needs further improvements.

---

> ### Author Response · Authors · 2022-11-10
> **Response to Reviewer XKxN**
>
> Thank you very much for your comments. We will illustrate your concerns as follows:
>
> **This work proposes to augment training data with the proposed QFA. Only the background information is replaced during the augmentation. However, the foreground information also varies significantly for complex environments. For instance, as shown in Figure 1, the targets in the source and target domains have different scales.**
>
> We acknowledge that the foreground information varies significantly. However, the targets in the source domain also have different scales, which means the model can learn the scale-invariant representations for the targets during training.
>
> Our approach builds on the observation that a collection of datasets is more likely to be in certain scenarios. For example, target object 1 in the source domain is collected in three environments of A, B, and C, and target object 1 in the target domain is collected in environments such as A, C, D, and E. Excessive training of the model on a fixed background will make it difficult to distinguish between the target object and the novel backgrounds when migrating to a new domain. The poor segmentation results of the baseline model in Figure 4 also demonstrate this. However, our method allows the model to continuously enhance the source data by replacing the background from other data in the source domain during training, and gradually expand the replaced background area as the training progresses. Therefore, the model can distinguish targets from backgrounds more robustly.
>
>
> **I am still confused about the detailed motivation for using AdaIN. It is originally used in style transfer, and this work seems to use it to mitigate the “style” of the query. However, AdaIN requires supervised training to present different styles, while the target domain data is not available during training for this task.**
>
> Yes, the AdaIN is originally used in style transfer. However, recent years have witnessed great success of AdaIN in single domain generalization tasks [1,2].
> Even within the same dataset, different samples will have different styles, the introduce of AdaIN can keep the content unchanged while randomize styles from other data. The results in Table 1 also demonstrate that introducing AdaIN in visual modality can increase the model’s generalization ability. In this paper, we observe that different query styles will lead to a deviation in the cross-attention map between the visual features and the query text when the model is migrated to the target domain. Thus the unimportant text information is integrated into the visual pixels’ representation, which will cause deviations in the segmentation results. Therefore, we design AdaIN on attention map to alleviate this problem.
>
> [1] Reducing Domain Gap by Reducing Style Bias
>
> [2] Semantic-Aware Domain Generalized Segmentation
>
> **The technical novelty of this paper is not up to the standard of ICLR. I believe that model performance is (marginally) improved but combining a data augmentation method and AdaIN and then applying them to the segmentation task is not appealing to me.**
>
> In this paper, we first give a detailed analysis of why the existing single-modality DG methods can not be directly applied to our proposed new task.
> Directly using these methods to enhance the two modalities separately and then fusing them together may suffer from negative effects, since it is difficult to ensure that the generalization directions of these two modalities are consistent.
> Based on these analyses, we design a novel progressive cross-modal background augmentation strategy, and constrain the synthetic domain $\bar{S}$ with contrastive learning. Unlike other DG methods where data augmentation is uncertain, our method can use the textual modality as a guide to orientate the augmentation of visual modality, and perform meaningful replacement augmentation on the visual background. As the area of visual background enhancement continues to expand during training, the constraint ability of contrastive module is also gradually enhanced, and the robustness of the model is also gradually improved.
>
> Our method is somewhat similar to the process of adversarial learning, but is more stable than it. To the best of our knowledge, our methods have not been used in any DG tasks. Our AM-AdaIN can introduce style randomization on cross-attention map to make the model learn a more robust relationship between query and visual feature.
> Besides, our analysis can also inspire subsequent works on multi-modal DG research.
>
> **The presentation should be improved. For instance, in the first sentence of the introduction, it seems that “byGavrilyuk” should be separated into two worlds by spacing. Similarly, “AdapINHuang” in the last paragraph of the introduction.**
>
> Thank you very much for pointing out these! We have revised these in our paper.

---

### Official Review · Reviewer_yPJV · 2022-10-24

**Confidence:** 4
**Correctness:** 3
**Technical Novelty And Significance:** 2
**Empirical Novelty And Significance:** 2
**Recommendation:** 6

**Clarity, Quality, Novelty And Reproducibility:**

The paper is well written and easy to follow. The novelty of this work is enough.

**Strength And Weaknesses:**

Strength:
1.	This work addresses multi-modal DG problem in query-based video segmentation task, which is still an untouched field. Multi-modal DG is meaningful and has its specific challenge. The authors’ effort on handling the multi-modal DG problem is encouraged.
2.	The authors observe that actions belonging to the same type of actors in different domains share similar representations in the query-video latent space, and the main reason for the performance degradation is that visual background and contextual complexity vary widely across different domains. Such observation is helpful.
3.	The proposed QFA and AM-AdaIN are reasonable.
4.	Extensive experiments validate the effectiveness of the proposed model.

Weakness:
1.	The performances are over-claimed that “From the results we can see that our method outperforms the baseline model by a large margin on all three generalization tasks, demonstrating the effectiveness of our proposed QFA and AM-AdaIN modules.” In 4.3 Main result. The performance of A2R in Table I is just comparable to existing methods.
2.	The proposed AM-AdaIN module is not illustrated in Fig. 2. It is hard to understand how it works in the proposed method.
3.	Can we replace the L_const into triplet loss, rather than Moco. Besides, the \tau in L_const is not addressed.
4.	Some typos:
a)	Bold “c ” in c_{hw}^k of \sigma(C_k)
b)	C’_K -> C’_k in Eqn. 9
c)	Inconsistence between A2R and A2D Sentences to RVOSpart in Table I and Table II.



**Summary Of The Paper:**

This work addresses the task of multi-modal DG where each modality has to cope with its own domain shift. Specifically, the specific scenario of query-based video segmentation is studied to better advance the generalization ability of the model in the multi-modal situation. The authors observe that actions belonging to the same type of actors in different domains share similar representations in the query-video latent space, and the main reason for the performance degradation is that visual background and contextual complexity vary widely across different domains. The authors propose a Query-guided Feature Augmentation (QFA) module that uses the attention scores between query and video frames to distinguish query-related foreground regions from unrelated background regions for synthesizing novel visual features. AdapIN is adopted to remove the impact of style in attention map during training, and gradually introduce statistics from other samples to help the model learn robust cross-modal relationships. Extensive experiments validate that the proposed model has better generalization ability.

**Summary Of The Review:**

Please see the Strength and Weakness.

---------

After reading the author response and other fellow reviewers' comments, I see that some concerns have been addressed, while others still remains unsolved. Based on the technical content and the author response, I decide to revise the score.

---

> ### Author Response · Authors · 2022-11-10
> **Response to Reviewer yPJV**
>
> Thank you very much for your acknowledgement. We will answer your questions as follows:
>
> **1. The performances are over-claimed that “From the results we can see that our method outperforms the baseline model by a large margin on all three generalization tasks, demonstrating the effectiveness of our proposed QFA and AM-AdaIN modules.” In 4.3 Main result. The performance of A2R in Table I is just comparable to existing methods.**
>
> Thank you for pointing out this. We have revised the corresponding description in our paper.
>
> **2. The proposed AM-AdaIN module is not illustrated in Fig. 2. It is hard to understand how it works in the proposed method.**
>
> Sorry about this. It is quite difficult to illustrate AM-AdaIN with figure, since it contains too many mathematical formulas. The main idea of our AM-AdaIN is, different query styles prefer different positions and channels in visual features, which can also be regarded as a style in a domain. Our AM-AdaIN can introduce style randomization on cross-attention map to make the model learn a more robust relationship between query and visual feature.
>
> **3. Can we replace the L_const into triplet loss, rather than Moco. Besides, the \tau in L_const is not addressed.**
>
> Thank you very much for your advice. Triplet loss is a feasible alternative method. We do a simple experiment and found that Triplet loss can also help constrain the augmented novel domain $\bar{S}$, but the convergence process is not as stable as Moco. Our analysis is that Moco contains a large number of negative samples, which allows the model to distinguish the difference between positive and negative samples faster.
>
> $\tau$ is a temperature hyper-parameter, here we keep it consistent with InfoNCE and Moco, and set it to 0.07.
>
> **4. Some typos: a) Bold “c ” in c_{hw}^k of \sigma(C_k) b) C’_K -> C’_k in Eqn. 9 c) Inconsistence between A2R and A2D Sentences to RVOSpart in Table I and Table II.**
>
> Thank you very much for pointing out these! We have revised these in our paper. Please refer to the latest paper.

---

### Official Review · Reviewer_NKmv · 2022-10-24

**Confidence:** 4
**Correctness:** 3
**Technical Novelty And Significance:** 2
**Empirical Novelty And Significance:** 2
**Recommendation:** 5

**Clarity, Quality, Novelty And Reproducibility:**

The task is novel, but the technical contribution is incremental, resulting limited novelty and originality. The proposed method is clearly presented in general. Some implementation details, especially how the baselines are implemented for multi-modal generalization setting, are not given. The code is not provided. I doubt exact reproducibility.

**Strength And Weaknesses:**

Strengths:
1. The studied domain generalization task, i.e., multi-modal generalization for query-based video segmentation, is novel and has not been investigated before. The idea is interesting.
2. Two modules are designed to address this challenging task, i.e. QFA and AM-AdaIN. This method achieves better results than the compared baslines. Ablation studies seem effective and sufficient.
3. The presentation is generally good.

Weaknesses:
1. Motivation is not clear enough. Some source-only (baseline) examples would better show the requirement of domain generalization for query-based video segmentation.
2. The technical contribution is relatively limited. Most components, e.g. both loss functions, are based on existing techniques.
3. I expected there are at leaset two source domains so that domain-invariant representations can be learned. I do not understand why only one source domain is used. This is also domain generalization but not the typical setting.
4. Baselines and some implementation details are not clear enough. For example, the compared baselines are mainly single-modal based methods. How to use them for the multi-modal generalization task?
5. The analysis on why the proposed method performs better in the experiment part is weak.
6.


**Summary Of The Paper:**

This paper studies cross-modal domain gereralization on the query-based video-segmentation task. Two modules, i.e., Query-guided Feature Augmentation (QFA) and attention map adaptive instance normalization (AM-AdaIN) are proposed for the multi-modal generalization task. Experiments are conducted on three domain generalization tasks for query-based video segmentation.

**Summary Of The Review:**

Novel domain generalization task but limited technical contrbution, good experimental results but insufficient analysis and insights behind the results, and generally fluent presentation.

---

> ### Author Response · Authors · 2022-11-10
> **Response to Reviewer NKmv**
>
>
> Thank you very much for your comments. We will illustrate your concerns as follows:
>
> **Motivation is not clear enough. Some source-only (baseline) examples would better show the requirement of domain generalization for query-based video segmentation.**
>
> Thank you very much for your advice, the source-only results are as follows:
>
> | Source only   | P@0.5 | P@0.6 | P@0.7 | P@0.8 | P@0.9 |
> |---------------|-------|-------|-------|-------|-------|
> | A2D Sentences | 65.20 | 59.31 | 49.09 | 32.36 | 8.34  |
> | RVOS          | 52.45 | 46.32 | 39.97 | 28.16 | 13.84 |
>
> | Target only   | P@0.5 | P@0.6 | P@0.7 | P@0.8 | P@0.9 |
> |---------------|-------|-------|-------|-------|-------|
> | R2A           | 48.65 | 40.95 | 30.27 | 16.11 | 2.29  |
> | A2R           | 38.46 | 30.61 | 21.00 | 10.29 | 1.96  |
>
> From the results we can see that the performances drop a lot when migrating to the new domain in both two datasets.
> Thus it is necessary to introduce domain generalization (DG) to increase the model’s generalization ability.
> We have added these descriptions in Figure 1, please refer to the latest paper.
>
> **The technical contribution is relatively limited. Most components, e.g. both loss functions, are based on existing techniques.**
>
> In this paper, we first give a detailed analysis about why the existing single-modality DG methods can not be directly applied in our proposed new task.
> Directly using these methods to enhance the two modalities separately and then fusing them together may suffer from negatively affects, since it is difficult to ensure that the generalization directions of these two modalities are consistent.
> Based on these analysis, we design a progressive cross-modal background augmentation strategy, and constrain the synthetic domain $\bar{S}$ with contrastive learning. Unlike other DG methods where data augmentation is uncertain, our method can use the textual modality as a guide to orientate the augmentation of visual modality, and perform meaningful replacement augmentation on the visual background.
>
> Our method is somewhat similar to the process of adversarial learning, but is more stable than it. As the area of visual background enhancement continues to expand during training, the robustness of the model is also gradually improved.
> To best of our knowledge, our methods have not been used in any DG tasks.
> Besides, our analysis can also inspire the subsequent works on multi-modal DG research.
>
> **I expected there are at least two source domains so that domain-invariant representations can be learned. I do not understand why only one source domain is used. This is also domain generalization but not the typical setting.**
>
> Single domain generalization is also a very important setting in DG tasks, some previous works have been proposed to investigate this problem [1-3].
> After all, in real-world conditions, it is very likely that we can only obtain one source domain to train our model, and there are many unknown test domains.
> Therefore, our works mainly study how to obtain a better robustness of the model when only a single source domain can be obtained.
>
> [1] Adversarially Adaptive Normalization for Single Domain Generalization
>
> [2] Progressive Domain Expansion Network for Single Domain Generalization
>
> [3] Learning to Learn Single Domain Generalization
>
> **Baselines and some implementation details are not clear enough. For example, the compared baselines are mainly single-modal based methods. How to use them for the multi-modal generalization task?**
>
> Thank you very much for pointing out this. The main implementation details are shown in Appendix A.2. However, as you mentioned, we also miss some important details. We use these baselines in visual morality to test their performances. Because we found that deploying them in visual modality works better than text modality. And if they are deployed in both two modalities, the performances are even worse. We have added these implementation details in Appendix A.2, please refer to the latest paper.
>
> **The analysis on why the proposed method performs better in the experiment part is weak.**
>
> The analysis is shown in the Appendix A.3 due to the page limit.
> Our proposed methods can effectively optimize the worst-case problem by generating a novel domain $\bar{S}$.
> At the initial training, $\bar{S}$ is close to original domain S. As the training progresses, β is gradually increasing, thus more background areas will be enhanced, the distance $D( S , \bar{S})$ is also getting larger.
> At the same time, the $L_{const}$ can guarantee the semantic consistency between the generated data and source data.
> Also our AM-AdaIN can introduce style randomization on cross-attention map to make the model learn a more robust relationship between query and visual feature.
>
> **As for code and reproductively.**
>
> The code, datasets and usage are now available in supplement material. We guarantee that the experiment results are absolutely reproducible.

---

### Official Review · Reviewer_HssY · 2022-10-25

**Confidence:** 4
**Clarity, Quality, Novelty And Reproducibility:** 1. Clarity
**Correctness:** 3
**Technical Novelty And Significance:** 2
**Empirical Novelty And Significance:** 2
**Recommendation:** 3

**Details Of Ethics Concerns:**

There are no Ethics Concerns.

**Strength And Weaknesses:**

Strength:
1. The research question is interesting.
2. The author did some experimental comparisons, and the results show that the proposed method has improved on some baselines.

Weaknesses:

1 The paper experiment is not sufficient.

-a. Does Table 1 compare other video segmentation methods? Does Table 1 compare with other domain adaption methods? Why does the title here compare with other generation methods?

-b. From Table 2, the improvement of each module is minimal. Combining Tables 1 and 2, I doubt whether the motivation for introducing domain adaption is reasonable. I'm concerned that the author is simply doing a combination of methods and ignoring the rationale of the research motivation.

2. Insufficient analysis of the rationality of domain adaption.

-a. Does the paper have visualization results of domain adaption? I want to know what domain adaption means specifically for video segmentation tasks.

3. The details of the method of the thesis are not clear.

-a. How is Equation 1 trained? How does it relate to other loss functions (Equations 6 and 11)?

-b. In Figure 2, what is the output of the augmented feature? Where did he enter the part of a?

-c. Is the domain adaption of this paper merely from data augmentation?




**Summary Of The Paper:**

1. The paper attempts to introduce domain generalization to solve query-based video segmentation.
2. The paper proposes QFA and AM-AdaIN modules to process query and video information, respectively.
3. The authors conduct experiments on the A2D Sentences, Refer-Youtube-VOS (RVOS), and J-HMDB Sentences datasets.

**Summary Of The Review:**

The paper lacks analysis and experimentation on research motivation. Some details of the method are unclear. I hope the author answers my question in the response. I'll change the score based on the author's reply.

---

> ### Author Response · Authors · 2022-11-10
> **Response to Reviewer HssY**
>
> Thank you very much for your comments. We will illustrate your concerns as follows:
>
> **The paper experiment is not sufficient:**
>
> A: Sorry about the confusion. In this paper, we mainly study how to improve the generalization ability of the model when it faces domain shifts in multimodal scenarios.
> Thus we compare our model with other state-of-the-art domain generalization methods.
> Also domain adaptation (DA) and domain generalization (DG) are two different tasks (the former can obtain target domain data during training, while the latter cannot), the performance of DA and DG will not be compared directly in most DG papers.
> FYI, we implement a traditional DA method Maximum Mean Discrepancy (MMD)[1] on visual modality on this task:
>
> | Task | P@0.5 | P@0.6 | P@0.7 | P@0.8 | P@0.9 |
> |------|-------|-------|-------|-------|-------|
> | A2R  | 34.16 | 28.62 | 22.83 | 14.57 | 4.43  |
> | R2A  | 47.36 | 40.62 | 32.82 | 20.59 | 4.35  |
>
> [1] Learning transferable features with deep adaptation networks
>
> B: The multi-modal domain generalization task is a very difficult task and has not been researched before.
>
> | Source only   | P@0.5 | P@0.6 | P@0.7 | P@0.8 | P@0.9 |
> |---------------|-------|-------|-------|-------|-------|
> | A2D Sentences | 65.20 | 59.31 | 49.09 | 32.36 | 8.34  |
> | RVOS          | 52.45 | 46.32 | 39.97 | 28.16 | 13.84 |
>
> | Target only   | P@0.5 | P@0.6 | P@0.7 | P@0.8 | P@0.9 |
> |---------------|-------|-------|-------|-------|-------|
> | R2A           | 48.65 | 40.95 | 30.27 | 16.11 | 2.29  |
> | A2R           | 38.46 | 30.61 | 21.00 | 10.29 | 1.96  |
>
> (Source only means the model is trained and tested on the same domain,
> while the target only means the model is trained on another domain, and directly transferred to this domain.)
>
> From the results we can see that the performances drop a lot when migrating to the new domain in both two datasets.
> It is necessary to introduce domain generalization to solve this problem because we can not guarantee that the data distribution is always the same.
> Traditional DG methods have been proved successful in single modality DG tasks, however, as shown in Table 1 (in the paper), they are all fail in multi-modal DG task.
> The research motivation of this paper is that multi-modality is a very common scenario in deep learning applications.
> We first analyze the reasons why traditional DG methods cannot work in multi-modal tasks, and then propose a novel multi-modal fusion DG strategy to effectively solve this new problem.
> Our method can also inspire subsequent works on multi-modal DG research.
>
> Compared with traditional single modality DG methods, our proposed method can effectively alleviate the multimodal domain shift problem, taking mean IoU as an example, our method can improve by 8.34%, 5.11%, and 6.08% on the three tasks, respectively.
> The results in Table 2 (in the paper) also demonstrate the effectiveness of our proposed modules.
>
> **Does the paper have visualization results of domain adaption? I want to know what domain adaption means specifically for video segmentation tasks.**
>
> In this paper, we mainly study domain generalization tasks, so we do not have visualization results of domain adaption.
> Figure 4 and Figure 5,6,7 in the appendix show the visualization results on three DG tasks, the model is trained on the first dataset and directly transferred to the second dataset to show the visualization results.
> DG in this task means that the model can still maintain a good segmentation result when visual and text modalities have to face their own domain shifts across different datasets.
>
> **The details of the method of the thesis are not clear.**
>
> A. Eq 1 is the theoretical analysis of the DG task. It means that we try to find a better parameter phi, which can maintain a good segmentation ability even in the worst case of the target domain. The L-seg in Eq 1 is Eq6. And we use Eq 11 to constrain the distance between augmented novel domain $\bar{S}$ and original domain S.
>
> B. The output of the augmented feature is the combined feature of the foreground in the source data and the background region in the augmented data. In this paper, we insert the query-guided feature augmentation module in the first QVIM layer, so the augmented feature is the visual feature $\hat{V_{1}}$ in part a.
>
> C. The domain generalization of this paper comes from two parts: the first one is query-guided feature augmentation, and the second one is adaptive instance normalization.

---

### Decision · Program_Chairs · 2023-01-20

**Decision:**

Reject

**Justification For Why Not Higher Score:**

technical contributions  limited

**Justification For Why Not Lower Score:**

NA

**Metareview: Summary, Strengths And Weaknesses:**

This paper studies domain generalization in query-based video segmentation, which aims to segment
the queried actors or objects in video based on the given natural language query. The reviewers find the task  novel. At the same time, they also feel that the technical contributions are limited.